# FSF-GA: A Feature Selection Framework for Phenotype Prediction Using Genetic Algorithms

**DOI:** 10.3390/genes14051059

**Published:** 2023-05-09

**Authors:** Mohammad Erfan Mowlaei, Xinghua Shi

**Affiliations:** Department of Computer and Information Sciences, Temple University, 925 N. 12th Street, Philadelphia, PA 19122, USA; mohammad.erfan.mowlaei@temple.edu

**Keywords:** genetic algorithm, machine learning, phenotype prediction, genomics

## Abstract

(1) *Background:* Phenotype prediction is a pivotal task in genetics in order to identify how genetic factors contribute to phenotypic differences. This field has seen extensive research, with numerous methods proposed for predicting phenotypes. Nevertheless, the intricate relationship between genotypes and complex phenotypes, including common diseases, has resulted in an ongoing challenge to accurately decipher the genetic contribution. (2) *Results:* In this study, we propose a novel feature selection framework for phenotype prediction utilizing a genetic algorithm (FSF-GA) that effectively reduces the feature space to identify genotypes contributing to phenotype prediction. We provide a comprehensive vignette of our method and conduct extensive experiments using a widely used yeast dataset. (3) *Conclusions:* Our experimental results show that our proposed FSF-GA method delivers comparable phenotype prediction performance as compared to baseline methods, while providing features selected for predicting phenotypes. These selected feature sets can be used to interpret the underlying genetic architecture that contributes to phenotypic variation.

## 1. Introduction

Recent advancements in genetic sequencing techniques have led to a significant increase in the volume of available genomic data over the past decade [1]. Access to vast quantities of genomic data necessitates the utilization of powerful data mining techniques and analytics to effectively extract relevant information. By uncovering key insights from such data, it becomes possible to identify genetic factors associated with human diseases, as well as to enhance breeding strategies in animals and plants [2]. Genomic and genetic data present challenges that hinder the efficacy of simple machine learning (ML) algorithms and statistical approaches. Genomic selection or phenotype prediction is one such challenge in genomic data analytics and involves predicting qualitative traits (e.g., eye color) or quantitative traits (e.g., height) based on genetic variations among individuals [3]. Along with non-genetic factors, trait values can also be influenced by genetic variants located in one or multiple locations in the genome [4]. These genetic variants can exhibit linear or non-linear interactions, which are, respectively, referred to as additive or epistatic effects, in determining the trait values [5]. Epistasis represents non-linear relationships among genetic variants and can extend beyond pairwise interactions and encompass higher-order interactions among multiple genetic variants [6]. Consequently, phenotype prediction in the presence of epistatic effects is a challenging combinatorial task, since modeling such interactions demands significantly complex models.

Another factor that makes phenotype prediction a challenge is the dimensionality of genomic data; this *curse of dimensionality* refers to the vast number of features that greatly outnumber the available samples [7]. Traditionally, approaches used to combat this problem using feature selection fall into four categories: *filters*, *wrappers*, *embedded* and *hybrid* approaches. Filters, such as information gain [8] and fisher score [9], mainly rely on statistical metrics in order to estimate correlation between features and phenotypes to mark the highest contributing features to phenotype prediction. Compared with wrappers, these approaches are usually several times faster; however, at the same time, they deliver less optimal results. Wrappers, on the other hand, employ kernels of ML models to perform Sequential Feature Selection (SSF) that can be either forward or backward [10]. Embedded methods are equipped with an internal mechanism based on techniques like regularization or penalty to identify features contributing to prediction during the training process [11], such as least absolute shrinkage and selection operator (Lasso) and Elastic Nets. Hybrid methods offer a promising solution by combining the strengths of both filters and wrappers to overcome their respective shortcomings, resulting in improved predictive accuracy [12,13].

Phenotype prediction is a rich field of study, featuring an array of established methods and techniques, including Linear Mixed Models (LMMs), statistical approaches, machine learning (ML) models and advanced Deep Learning (DL) frameworks. LMMs are a popular choice for genomic selection and phenotype prediction. Some well-known LMMs widely used in practical genomic selection practices are Best Linear Unbiased Prediction (BLUP) and its extensions including ridge regression BLUP (rrBLUP) [14], genomic relationship BLUP (GBLUP) [15] and single-step genomic BLUP (ssGBLUP) [16]. Such models predict trait values using linear functions, assuming that the effect of each Single Nucleotide Polymorphism (SNP) on phenotypes is drawn from a normal distribution with the same variance. Statistical approaches, such as Lasso [17] and Elastic Nets [18], have been proposed for phenotype prediction, which can simultaneously conduct feature selection and phenotype prediction. These methods, however, had room for improvement in genetics-related problems. As such, a number of variants for these methods were proposed in order to overcome the limitations of the original approaches; to name but a few: Spike-and-Slab Lasso Generalized Linear Model (ssLasso GLMs) [19], Empirical Bayesian Elastic Net (EBEN) [20], its parallel version (parEBEN) [21], Multiple-trait Bayesian Lasso (MBL) [22], Bayesian Ridge Regression [23] and generalized Poisson regression (GPR) [24].

Classical ML models used for phenotype prediction include random forest (RF) [25] and support vector machine (SVM) [26]. Currently, there is a growing number of studies utilizing cutting-edge DL for phenotype prediction since DL has shown superior prediction performance in many other domains such as computer vision and text mining. For example, DeepGS [27,28] use saliency maps or layer activity to determine an SNP’s contribution to a particular phenotype. Although these ML and DL models for phenotype prediction provide desirable performance for phenotype prediction, they cannot provide feature selection and thus do not easily convey the genetic variants that contribute to phenotypic variation.

Evolutionary and swarm intelligence algorithms, including Genetic Algorithms (GAs), represent another promising category of methods that excel at identifying informative features for phenotype prediction. As a result, GA has become a popular tool for genetic studies aimed at addressing a diverse range of challenges, including phenotype prediction [12,13,29,30,31] and epistasis detection [32,33,34].

With the exception of [28], the methods described above are tailored specifically to classification tasks in phenotype prediction and cannot be readily applied to quantitative trait locus (QTL) analysis. Predicting quantitative values with GA presents a unique set of challenges, including the difficulty of identifying a suitable fitness function for evaluating GA performance, given that commonly used metrics such as Mean Squared Error (MSE) and $R^{2}$ may yield suboptimal results. Moreover, the complexity of relationships among genetic features, which may include interactions, redundancy, correlation or epistasis, coupled with the sheer number of such features, can make it prohibitively difficult to identify convergent and efficient solutions for a GA.

Hence, to allow phenotype prediction of quantitative traits, we have developed a novel framework called Feature Selection Framework for Phenotype Prediction (FSF-GA) that uses a hybrid method to tackle the two aforementioned challenges. FSF-GA allows for phenotype prediction as well as feature selection for QTL detection based on GA that explores relationships among genetic features. In summary, FSF-GA consists of two primary stages that support feature selection in synergy with a regression model for phenotype prediction. In the first stage, we perform pre-processing in order to reduce the search space of the features and in the second stage, we utilize GA to find the best combination of SNPs to predict a quantitative trait under investigation.

## 2. Results

### 2.1. FSF-GA Implementation

FSF-GA is designed to address both phenotype prediction and QTL detection problems with consideration of correlations or relationships among genetic features such as Linkage Disequilibrium (LD) blocks. In order to evaluate FSF-GA in our experiments, the dataset (a yeast dataset is used in this study) is first shuffled and separated into training and test sets using a fixed random seed, as illustrated in Figure 1; this is a normal course of action for evaluating an ML approach. Next, we perform feature selection on both sets using Pearson’s Correlation Coefficient (PCC) between SNPs and the target phenotype. Afterwards, GA is applied to the training set in order to find the optimal set of features that results in the best $R_{\mathrm{Adj}}^{2}$ between the predicted and observed phenotypes. Finally, a test set is used in order to evaluate the performance of the GA-based regression model in FSF-GA.

### 2.2. LD Concordance

LD cutoff used in the pre-processing step of Algorithm 1 affects detected loci. More specifically, the higher we set the LD cutoff, the more variation, in terms of cardinality and concordance, we observe in QTL sets generated as the final output of the GA. This outcome is presented in Table 1, where the statistics of the yeast dataset after pre-processing (third column) and running the GA three times (last column) are reported. Therefore, in order to keep the results steady, we run the GA multiple times for each setting and consider the intersection of resulting sets as the final set of associated loci with a quantitative trait. Similarly, phenotype prediction performance of FSF-GA is evaluated using the intersection sets.
**Algorithm 1:** Pseudo-Code for the Pre-Processing Function
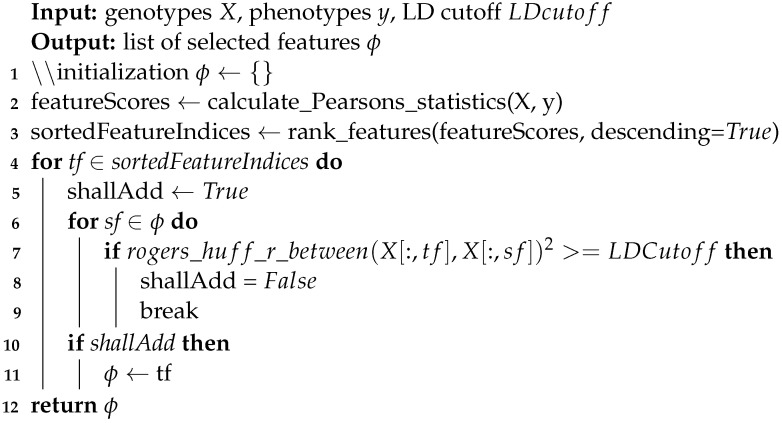


To compare our findings with previous publications, we used LD concordance between our results and those of [5], which were acquired using a mixed model for repeated measures (MMRM). To calculate LD concordance, for each trait, we looped through QTLs reported by [5] and saved the maximum LD score among each of those and SNPs detected via FSF-GA for the same trait. Results of this analysis are presented in Figure 2 as violin plots. We observe that, for the majority of the traits, LD concordance is high, especially for LD cutoff thresholds of 0.4 and 0.5. A high LD concordance indicates that the FSF-GA pipeline can identify genetic loci that are either QTLs or in high LD with previously reported QTLs.

### 2.3. QTL Detection

Though FSF-GA is not designed to effectively mark QTLs at the current stage, we observe that a fraction of SNPs reported in [5] directly overlap with our sets. For each trait, we used an overlap between intersections that resulted from six LD cutoff thresholds for FSF-GA and then calculated the direct intersection between our method and that of [5]. It is noteworthy that, as expected, many of these overlaps were mentioned as additive QTLs in [5]. These overlaps are listed, per phenotype, in Table 2.

### 2.4. Phenotype Prediction

For evaluating phenotype prediction of FSF-GA, we use mean absolute error (MAE), MSE and PCC to benchmark FSF-GA against other baseline methods, including classical ML models (RF and SVM) and genetic selection methods (rrBLUP and BGLR). For the ML models, such as RF and SVM, we train them on remaining features after the pre-processing step in our pipeline. Reported results for these models are the best out of each six LD cutoffs (0.2, 0.3, 0.4, 0.5, 0.6 and 0.7) per metric, which are considerably better than running them on the full set of features. For the RF regressor, after fine-tuning the model, number of trees and maximum depth of each tree are set to ceil(p/7) (where p is the number of features) and 7, respectively. In order to make the results fair, we run RF using 10 random splits and report the average performance metrics on the test set.

Regarding classical genetic selection models such as rrBLUP and BGLR [35], they are provided with the complete set of features. The results of this benchmark are illustrated in Figure 3 and Figure 4. The outcomes suggest that FSF-GA performs as well as rrBLUP and BGLR. FSF-GA exhibits comparable performance to SVM and RF. However, RF’s predictive performance can be highly variable in certain scenarios compared to other methods, such as Diamide/Indolacetic Acid, where RF’s predictions can be significantly better or worse than those of the other methods.

However, the performance of models can be affected by the selected data partition. In order to assert that FSF-GA performs comparably to rrBLUP and BGLR in terms of MSE, we perform 5-fold cross-validation on the first three traits, namely *Cobalt Chloride*, *Copper Sulfate* and *Diamide*. The results of this experiment are presented in Table 3 and Table 4. We can observe that FSF-GA does not perform as well as the other two in the case of *Diamide*. However, for *Cobalt Chloride* and *Copper Sulfate* there is at least one LD cutoff threshold in which FSF-GA performs as well as rrBLUP and BGLR.

## 3. Discussion

We observe that the majority of SNPs reported by previous research show a high LD concordance to those found via FSF-GA. In this regard, since there is no considerable difference in performance metrics such as MSE, the problem turns into a trade-off between LD concordance and I/U. The former is a measure of consistency between the results of FSF-GA with the previous research, while the latter is a measure of robustness of each individual run. Therefore, increasing the LD cutoff threshold reduces I/U and, at the same time, increases LD concordance. Overall, LD concordance is higher at LD cutoff thresholds above 0.4. LD cutoff thresholds of 0.4 and 0.5 show a reasonable amount of both I/U and LD concordance, making them preferable to other cutoff thresholds for individual runs. Of note, since increasing the LD cutoff threshold further than 0.5 does not result in tangible improvements in performance metrics, we abstain from checking LD cutoff values higher than 0.7. The discordance in results can be attributed to not using the full dataset by FSF-GA and/or the use of different LD blocks for prediction via the two methods. While the latter needs further investigation, which can be a focus of our future studies, it is not unexpected. In stepwise methods used for QTL detection, such as the one used in [5], the order of SNP selection can affect the process; i.e., the first few detected QTLs can lead to or prevent the selection of specific QTLs, which can prevent the model from selecting the optimal set of the QTLs. This shortcoming, theoretically, is not applicable to metaheuristic models such as FSF-GA, since selected loci can be removed and replaced during the process by the modification operators, such as mutation and crossover in the GA.

In the case of phenotype prediction, SVM performs worst, but the difference between the methods is not substantial. RF does better for some traits compared to others by a narrow margin. Since RF uses the same set of features that are fed into our GA, we suspect that the difference between our results and those of RF lies in the difference between the prediction power of Bayesian Ridge, which is used in FSF-GA as the estimator, and RF. Some models, such as neural networks and tree-based models, have the capacity to capture non-linear interactions between SNPs. Specifically, neural networks are suitable tools for detecting epistasis since they utilize non-linear activation functions in each layer. In contrast, Bayesian Ridge does not offer the same functionality; however, using other regressor models in the objective function of our GA adds significant computation load, so we avoid it in this study due to limitations in our hardware. Regardless, FSF-GA delivers results comparable to those of rrBLUP and BGLR, which are considered formidable baselines because in the yeast dataset the majority of contributions to trait variation result from additive effects [5]. Therefore, we expect that using neural networks instead of Bayesian Ridge can lead to a boost in prediction power and better epistatic QTL detection in FSF-GA, subjected to availability of more computational power which will be explored in future work.

The findings suggest that the LD cutoff used during the pre-processing step is a key factor in regulating the I/U ratio of identified SNPs. Moreover, it also impacts the stability of the SNP sets obtained for various traits and has a marginal influence on the accuracy of phenotype prediction. The reason is that when LD cutoff is higher, more correlated features are introduced to the search space and the SNP selection progress is influenced by this variation, while the presence of more SNPs increases the chance that a better QTL combination is found. In our current framework, Bayesian Ridge cannot capture complex QTL-QTL interactions; therefore, the performance gap using different cutoffs is insignificant. However, if we use other models capable of capturing such interactions, we have a trade-off between precision of QTL detection and phenotype prediction power.

## 4. Conclusions

In this study, we propose FSF-GA as a bi-objective framework for quantitative trait prediction and QTL detection. To the best of our knowledge, our method is the first method to make use of evolutionary algorithms for such a task. The advantage of metaheuristic approaches, such as FSF-GA, over stepwise models is their potential for breaking out of a local optimum and not being affected by initially selected QTLs in the process. A practical application of FSF-GA is to combine it with CRISPR genetic editing tools and/or base editing tools for targeted SNP editing and investigating phenotypic changes. This approach could provide precise and targeted SNP studies compared to traditional QTL mapping techniques.

For model evaluation, a widely used yeast dataset is used to benchmark our proposed method against other well-known models for quantitative phenotype prediction. Experimental results indicate that FSF-GA performs comparably to the baselines for phenotype prediction, while it can detect trait-associated LD blocks using pre-processed SNPs. Furthermore, our analysis reveals that QTLs detected by our framework show a reasonable LD concordance with the QTLs identified in previous research. Statistics show that the LD cutoff, in our proposed pre-processing step, affects phenotype predictive power and convergence of the GA in our framework. This LD cutoff is a trade-off between the convergence of the GA and phenotype predictive power of our method.

While FSF-GA shows reasonable performance in designated tasks, there is room for further improvement. We believe that using advanced models, such as neural networks, can lead to increased prediction performance and better QTL detection, given that powerful hardware is available. In our future work, we intend to expand this method in order to mark the exact QTLs in LD blocks that are responsible for predicting each quantitative trait, which, subsequently, improves phenotype prediction as well. Furthermore, our algorithm can be redesigned in a parallel fashion to reduce execution time, which can be another focus of our future work.

## 5. Materials and Methods

### 5.1. Dataset

In this study, we apply our method to a well characterized yeast Saccharomyces cerevisiae dataset [5]. The dataset contains sequenced genotype profiles of 4390 samples with 28,220 unique genetic variants (SNPs). The samples in the dataset are crosses between a laboratory strain (BY) and an isolate form of vineyard (RM), encoded as −1 and 1, respectively. In addition to the genotypes, the dataset contains phenotypes of the samples for 20 different growth traits. An important aspect of the dataset is the existence of epistasis, otherwise referred to as nonlinear gene–gene interactions, that contributes to the complexity of model training for trait prediction [36].

In order to evaluate our proposed method, we compare its results with baseline methods on ten phenotypes in this dataset. The dataset is split into two separate subsets, using a unique random seed, for training and test purposes, consisting of 90% and 10% of the samples, respectively. The aforementioned steps are identically repeated for baseline methods as well. Additionally, we perform a 5-fold cross validation on the first three traits in this dataset to ensure that the data split does not have a considerable effect on the model performance.

### 5.2. Feature Selection Framework for Phenotype Prediction Using Genetic Algorithm

Phenotype prediction involves solving two problems, namely epistatic interactions among loci and the curse of dimensionality. To address the latter, we propose FSF-GA in order to reduce the search space for effective SNPs in phenotype prediction. For the first problem, while our method does not directly address the epistasis detection problem, it can be used as a prior step in order to detect high-order epistasis. The overall pipeline for the proposed method in this study is illustrated in Figure 1. As a side note, henceforth, we use loci and features interchangeably.

The dataset is first partitioned into training and test sets. Afterwards, pre-processing is applied to both sets, restricting features that are used for the GA in the next step. In the GA, we aim to find the optimal set of features that maximizes our criteria, namely $R_{\mathrm{Adj}}^{2}$, on the training set. A regression model is then fit on the training set using the selected subset of features and we evaluate the performance using this model on the test set. Selected features can depend on the regression model used in the fitness function of the GA; however, the final output of the GA is only the set of features. Since GA is a stochastic algorithm, we run it more than once and consider the overlap of produced sets as the final output.

#### 5.2.1. The Optimization Problem

GA is a nature-inspired method and a major constitution of Computational Intelligence (CI), designed to solve real world problems [37]. Our objective in the GA is to find the optimal solution for an optimization problem.

In this study, given a certain regression model *M* and a dataset *D*, we look for the minimal subset of SNPs in *D* that provides us the best phenotype prediction results. The normal procedure for stopping a GA, in case the optimal goal is not met, includes (but is not limited to) setting a time limit on the runtime of the algorithm or the number of iterations. Here, we employ the latter method and set the maximum number of iterations to 5000.

There are a limited number of metrics used for regression problems. Among them, MSE is commonly used as a measure to compare different methods. However, through empirical study we found that $R_{\mathrm{Adj}}^{2}$, which has been used in the literature for regression problems [38], serves as a better objective function for the task at hand. This metric can be calculated as follows:(1)$$R_{\mathrm{Adj}}^{2}=1-\frac{\left(1-R^{2}\right)\left(n-1\right)}{n-p-1};$$where *p* is number of independent features selected for training the model, *n* is the number of samples and $R^{2}$ is calculated as below:(2)$$R^{2}=1-\frac{RSS}{TSS};$$where *RSS* is the sum of squares of residuals and *TSS* is the total sum of squares for a given trait. $R_{\mathrm{Adj}}^{2}$ and $R^{2}$ range from 0 to 1, where 1 is the best and 0 is the worst value. We set maximizing $R_{\mathrm{Adj}}^{2}$ and minimizing the number of features as the primary and secondary objectives, respectively. In other words, our optimization algorithm maximizes $R_{\mathrm{Adj}}^{2}$ of phenotype prediction using as few features as possible.

#### 5.2.2. Pre-Processing

The purpose of a GA in our approach is to find the most compact set of features, for each trait, that delivers the best predictive power. However, evolutionary algorithms alone cannot prioritize suitable features, resulting in extremely long run-times until convergence. Furthermore, genetic features are highly correlated due to LD structures or gene co-expression/regulation, which highlights the need for feature selection, similar to [21,39,40,41]. Therefore, in order to guide our GA, we first mark valid SNPs for each trait and our GA is only allowed to use them in order to form the output set of the features. To do so, we make use of PCC among each SNP and the target trait in order to sort the SNPs, based on their importance, in a decreasing fashion. Next, we calculate LD between SNP pairs in order to filter sorted SNPs, as demonstrated in Algorithm 1. LD between SNP pairs is calculated using *scikit-allel* package in Python.

#### 5.2.3. The Genetic Algorithm

In this section, we present the details of our GA. The inputs to our GA are the training set and valid feature indices acquired in the pre-processing step. The output is the set of selected SNPs that give the optimal result for predicting the phenotype of interest. Since randomness in evolutionary algorithms is inevitable, especially in this problem, we run the GA three times and use the intersection of outputs as the final set, for each setting.

The fundamental components of the GA are chromosomes and three functions named *fitness*, *mutate* and *crossover*. The overall process of the proposed GA is illustrated in Figure 5. In our algorithm, each chromosome contains a vector of binary values (1/0) called *genes*. In other words, genes refer to the parameters of the solutions in our problem, which are different from genes in genetics and through this paper, we use gene(s) only in the context of GA. The length of each array is equal to the number of loci in genotypes. Setting each element in these arrays to 1/0 indicates that the corresponding feature should be used/discarded in the respective data subset. In other terms, these arrays mask the presence of features in the dataset, as demonstrated in Figure 6. The *fitness* function in the proposed GA simply calculates the *fitness score* on the training set, that is, $R_{\mathrm{Adj}}^{2}$ in this study, using Bayesian Ridge regressor implementation from *Scikit-learn* package [42]. The key to selecting the model for the *fitness* function is that it should not have inherent *L1* penalty (e.g., Lasso), so that redundant features affecting model performance are removed in the process. Tabu Search (TS) [43] is incorporated into our GA in order to improve local search and prohibit it from re-checking previously visited solutions. Furthermore, TS can save time by preventing redundant calculations in the *fitness* function.

The *mutate* function takes a chromosome and modifies its genes, exploring the search space for the global optimum. Algorithm 2 contains the pseudo-code for the *mutate* function. The inputs of the *crossover* function are two chromosomes, named parents ($G_{P}$, $G_{D}$), their respective fitness scores and the *fitness* function. Generally speaking, in GA, a crossover combines two sets of genes, resulting in a new chromosome, named child ($G_{C}$), in which genes are inherited from either of the parents—performing exploitation and leading to convergence in search subspace. The same is applied in our *crossover* function. The pseudo-code of the *crossover* function is presented in Algorithm 3.

The base code of our GA is adopted from [44]. However, as mentioned above, the code was heavily modified. The size of the parent pool in our GA is set to 10. The rate of mutation and crossover is set dynamically according to the last three improvements. However, we have designed the algorithm so that the mutation/crossover rate cannot fall beneath 20% and in each turn, only one of these operations is performed on each chromosome. For example, if two out of three of the last improvements are achieved through crossover, then succeeding functions to apply on the next chromosomes, until a subsequent improvement is obtained, are crossover/mutation with a probability of 60/40%.
**Algorithm 2:** Pseudo-Code for the Mutate Function
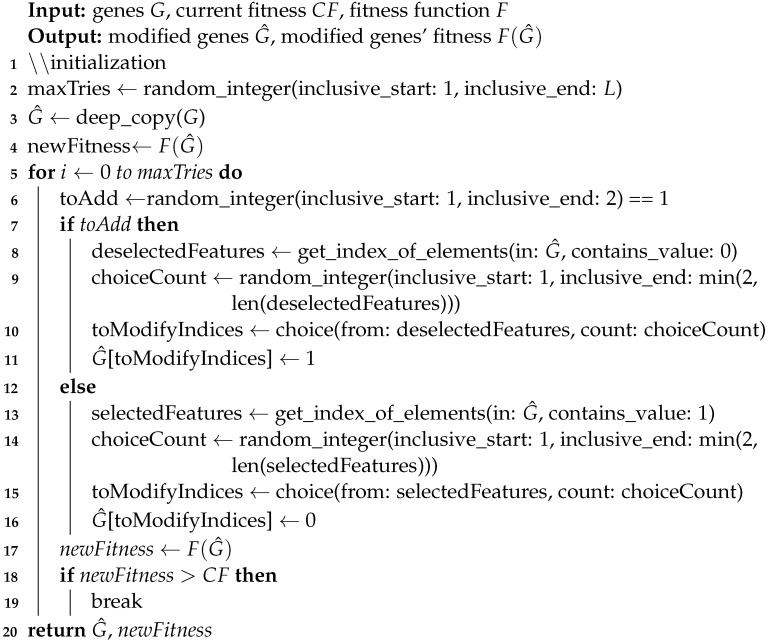


**Algorithm 3:** Pseudo-Code for the Crossover Function

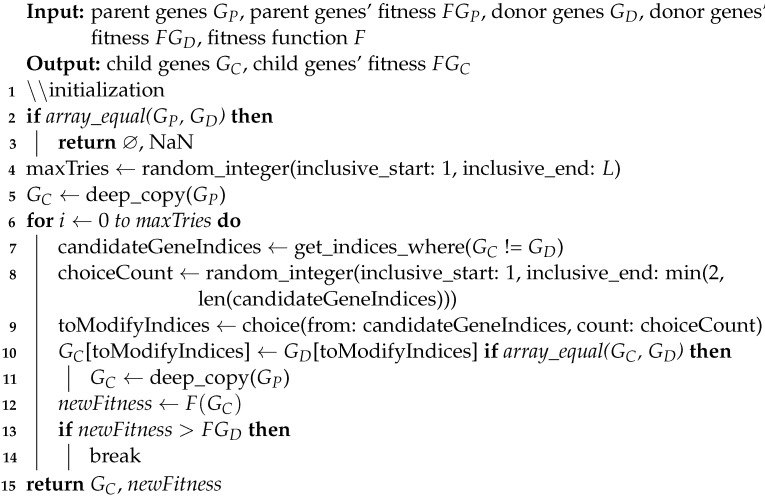



#### 5.2.4. Computational Complexity

The performance bottleneck of the proposed GA is the *fitness* function. The most expensive operation within this function involves training a Bayesian Ridge model on the data. Considering we have *n* samples and *p* features, the training takes $O(np^{2}+p^{3})$ operations. After the pre-processing step n≫p we can safely assume the cost of Bayesian Ridge to be $O(p^{2}n)$. Mutation and crossover take $O(Lnp^{2})$ since their inner loops (Lines 5 and 6) involve calls to the *fitness* function up to *L* times. The GA runs for *K* iterations and the parent pool holds *S* chromosomes in total; hence, the total computational complexity of our GA, in this study is $O(LKSnp^{2})$.

## Figures and Tables

**Figure 1 genes-14-01059-f001:**
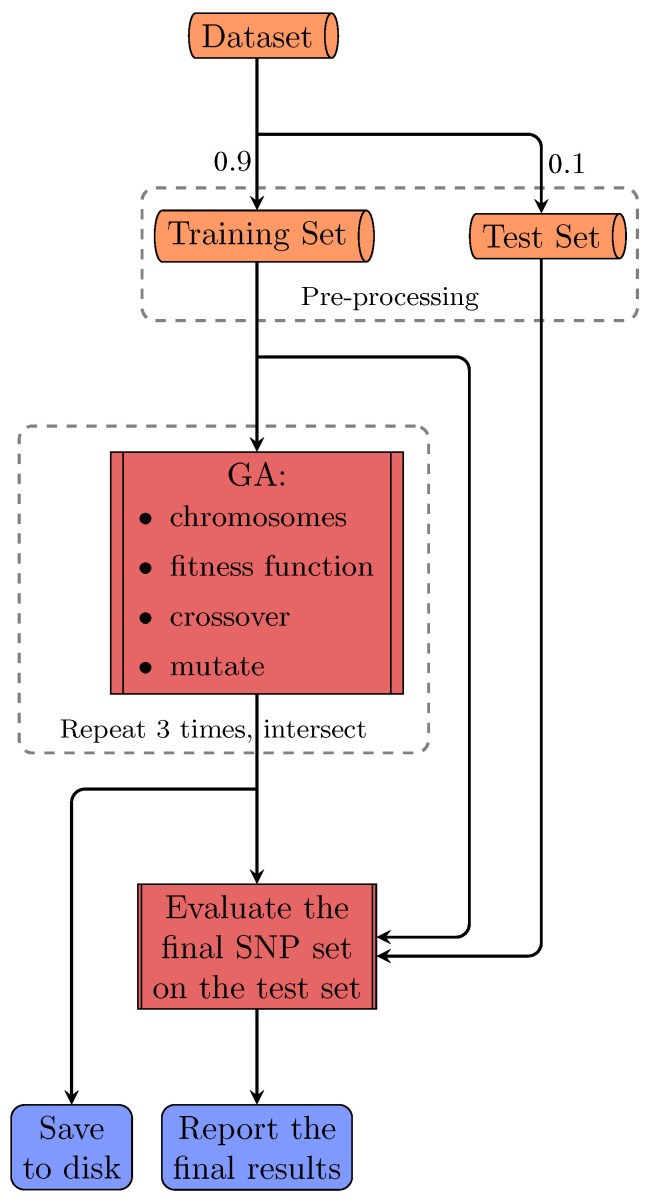
The overall pipeline of FSF-GA for a single run. The dataset is split into training and test sets. Pre-processing is then performed on both sets using the information extracted from the training set. Next, GA is applied on the training set three times. The intersection of the results from these three executions is considered as the final set of QTLs and used for the test set evaluation.

**Figure 2 genes-14-01059-f002:**
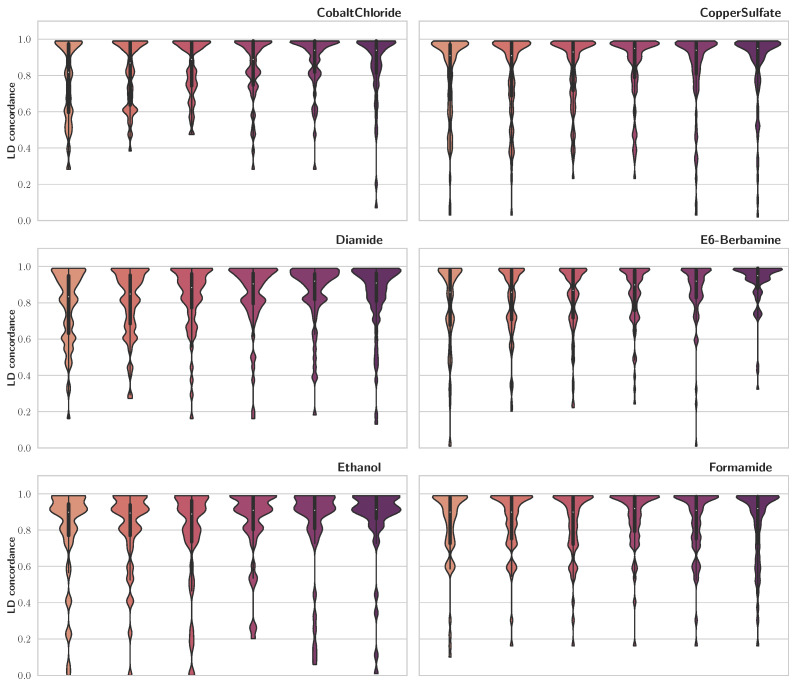
LD concordance between detected QTLs using FSF-GA and QTLs identified in [5]. Higher width indicates greater density. The central line in violin plots depicts quadrilles of the data, with the white circle as the median. We observe that, generally, higher LD cutoff thresholds result in more concordance between the two sets across different traits.

**Figure 3 genes-14-01059-f003:**
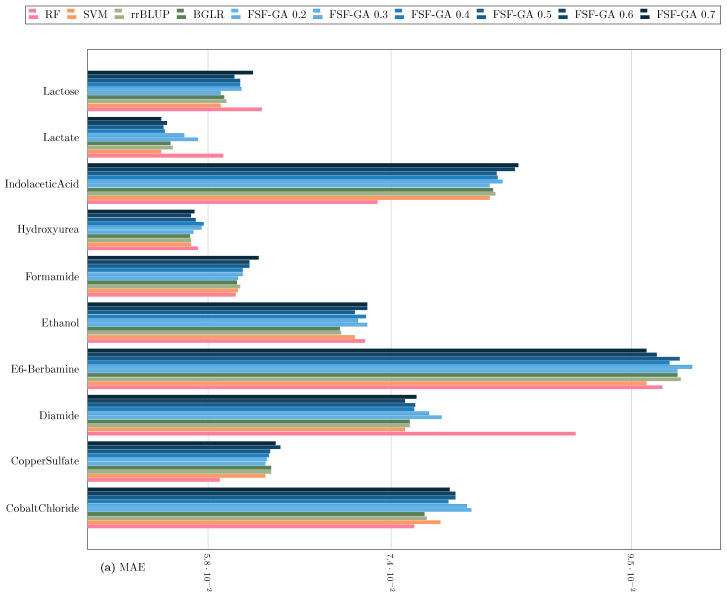
Performance Comparison of FSF-GA to baseline methods based on (**a**) MAE and (**b**) MSE (lower values indicate better results) with a one-time evaluation on a single split of the dataset. α in FSF-GA α corresponds to the LD cutoff threshold used in the pre-processing step. RF and SVM are applied to features for each of six LD cutoff thresholds in the pre-processing step and the best results among them, for each metric, are considered.

**Figure 4 genes-14-01059-f004:**
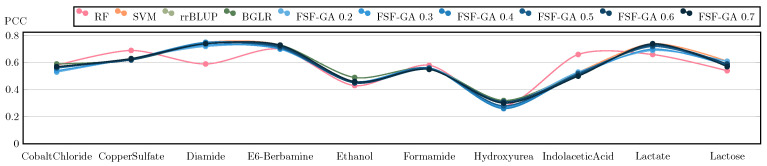
Performance Comparison of FSF-GA to baselines based on PCC (higher values indicate better results; best performance at +1). α in FSF-GA α corresponds to the LD cutoff threshold used in the pre-processing step. RF and SVM are applied to features for each of six LD cutoff thresholds in the pre-processing step and the best results among them, for each metric, are considered. We observe that based on PCC, FSF-GA performs similar to baseline methods.

**Figure 5 genes-14-01059-f005:**
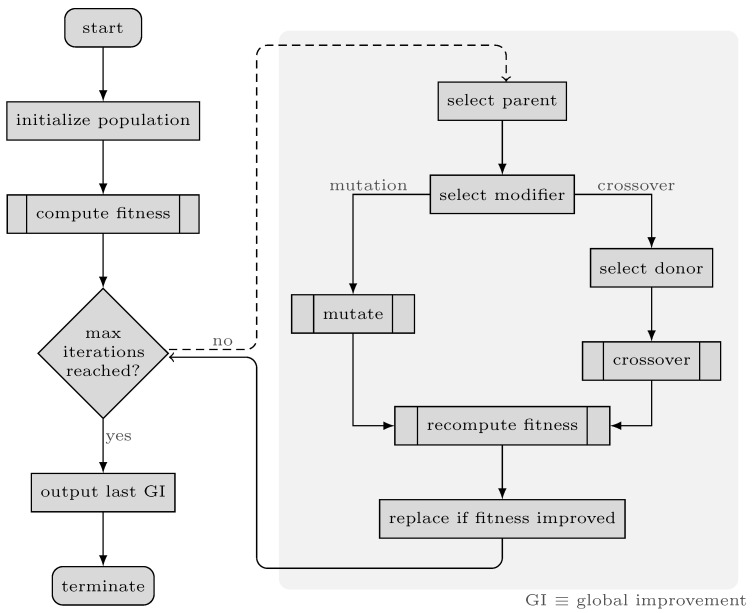
Flowchart of the genetic algorithm.

**Figure 6 genes-14-01059-f006:**
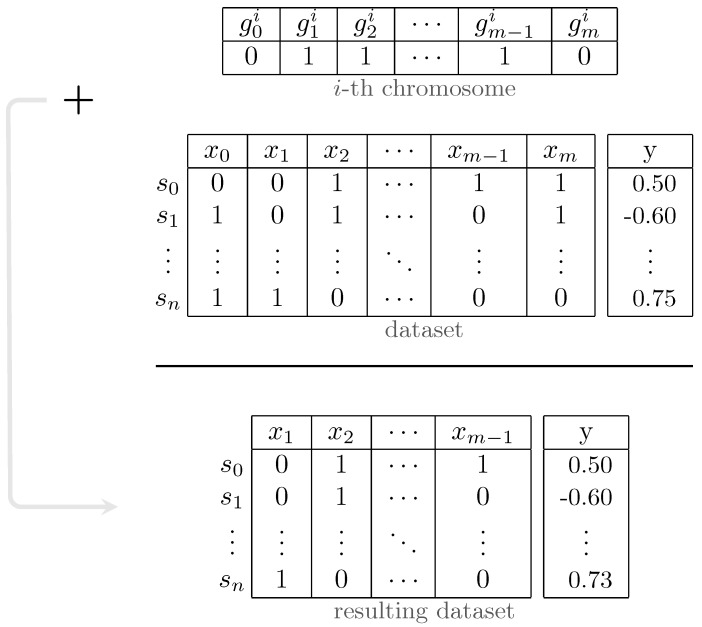
Process of masking features using chromosome’s genes in genetic algorithm.

**Table 1 genes-14-01059-t001:** SNP Statistics after pre-processing and running the GA three times on the yeast Saccharomyces cerevisiae dataset. *LD cutoff* refers to the LD threshold used in the pre-processing step. *Intersection*, in the last column, indicates the overlap between three runs on each LD cutoff and I/U indicates the intersection-to-union ratio. Higher values of I/U are desirable and indicate that the results have more concordance.

Trait	LD Cutoff	Remaining SNPs after Pre-Processing	Intersections (I/U)
Cobalt Chloride	0.2	122	87 (1.0)
	0.4	196	110 (0.96)
	0.7	449	136 (0.7)
Copper Sulfate	0.2	127	73 (1.0)
	0.4	201	102 (0.94)
	0.7	440	125 (0.73)
Diamide	0.2	127	86 (1.0)
	0.4	196	113 (0.92)
	0.7	450	150 (0.72)
E6-Berbamine	0.2	121	87 (1.0)
	0.4	194	119 (0.91)
	0.7	443	122154 (0.73)
Ethanol	0.2	123	72 (1.0)
	0.4	193	78 (0.85)
	0.7	444	114 (0.83)
Formamide	0.2	120	83 (0.94)
	0.4	193	106 (0.95)
	0.7	454	146 (0.68)
Hydroxyurea	0.2	117	76 (1.0)
	0.4	193	85 (0.93)
	0.7	436	65(0.39)
IndolaceticAcid	0.2	125	79 (0.98)
	0.4	197	83 (0.9)
	0.7	434	115 (0.68)
Lactate	0.2	122	78 (0.96)
	0.4	203	117 (0.97)
	0.7	439	152 (0.74)
Lactose	0.2	123	83 (1.0)
	0.4	204	104 (0.89)
	0.7	441	144 (0.75)

**Table 2 genes-14-01059-t002:** Shared QTL SNPs detected via FSF-GA and [5]. Items under *overlap positions* are represented as *chrN_P*, where N and P indicate chromosome number and position on the chromosome, respectively. These overlaps show that our method detects relevant features as QTLs.

Trait	Overlapping SNPs
Cobalt Chloride	chrIII_95776, chrV_290603, chrVII_375494, chrVII_609644, chrX_656567, chrXII_657536, chrXIII_13364, chrXVI_215587
Copper Sulfate	chrIV_605558, chrVIII_209176, chrIX_102310, chrXI_617158, chrXII_649837, chrXIII_50109, chrXV_558953
Diamide	chrVIII_114337, chrXII_53269
E6-Berbamine	chrI_193500, chrIV_424905, chrIV_614738, chrIX_261221, chrXIV_466588, chrXVI_813147
Ethanol	chrII_393269, chrIV_548896, chrXI_226101, chrIV_1174544, chrXIII_50109, chrXV_461793
Formamide	chrV_451448, chrVII_300190, chrVIII_114144, chrVIII_518945, chrXI_46843, chrXI_138463, chrXV_559465
Hydroxyurea	chrX_60299, chrXIII_862695, chrXV_34772, chrXV_182946, chrXV_580357
Indolacetic Acid	chrVIII_114144, chrXIII_410320
Lactate	chrI_39105, chrV_287674, chrXI_22343, chrXII_755553, chrXV_173080
Lactose	chrIV_639037, chrV_526092, chrVI_54494, chrXII_570032, chrXIII_25025, chrXIII_844047, chrXV_173340

**Table 3 genes-14-01059-t003:** Performance comparison of FSF-GA to rrBLUP and BGLR in a 5-fold cross-validation based on average MSE (lower values indicate better results). The values in the parentheses indicate standard deviation. We can observe that there is not a considerable performance gap between FSF-GA and baseline methods.

		Phenotype		
Method	LD Cutoff	Cobalt Chloride	Copper Sulfate	Diamide
rrBLUP	N/A	1.08 × 10^−2^ (4.09 × 10^−4^)	6.61 × 10^−3^ (4.74 × 10^−4^)	9.11 × 10^−3^ (8.68 × 10^−4^)
BGLR	N/A	1.08 × 10^−2^ (3.94 × 10^−4^)	6.59 × 10^−3^ (4.78 × 10^−4^)	9.09 × 10^−3^ (8.65 × 10^−4^)
FSF-GA	0.2	1.11 × 10^−2^ (3.31 × 10^−4^)	6.65 × 10^−3^ (4.83 × 10^−4^)	1.02 × 10^−2^ (8.17 × 10^−4^)
	0.3	1.12 × 10^−2^ (3.52 × 10^−4^)	6.66 × 10^−3^ (4.84 × 10^−4^)	9.96 × 10^−3^ (7.07 × 10^−4^)
	0.4	1.12 × 10^−2^ (3.76 × 10^−4^)	6.62 × 10^−3^ (4.70 × 10^−4^)	9.65 × 10^−3^ (6.39 × 10^−4^)
	0.5	1.11 × 10^−2^ (3.95 × 10^−4^)	6.71 × 10^−3^ (3.61 × 10^−4^)	9.63 × 10^−3^ (7.87 × 10^−4^)
	0.6	1.09 × 10^−2^ (2.29 × 10^−4^)	6.70 × 10^−3^ (3.35 × 10^−4^)	9.65 × 10^−3^ (7.80 × 10^−4^)
	0.7	1.11 × 10^−2^ (3.75 × 10^−4^)	6.76 × 10^−3^ (4.40 × 10^−4^)	9.66 × 10^−3^ (7.67 × 10^−4^)

**Table 4 genes-14-01059-t004:** Comparison of FSF-GA to rrBLUP and BGLR based on paired two-tailed *t*-test results over MSE values of each fold in a 5-fold cross-validation. A *p*-value less than 0.05 indicates that either rrBLUP or BGLR performs better or worse than FSF-GA, depending on the direction of the t-statistic (positive for better performance; negative for worse performance). Conversely, if the *p*-value is greater than 0.05, no single method dominates the other.

		rrBLUP		BGLR	
Trait	LD Cutoff	*t*-Statistics	*p*-Value	*t*-Statistics	*p*-Value
Cobalt Chloride	0.2	2.99	0.04	3.31	0.03
	0.3	2.24	0.03	3.82	0.02
	0.4	4.013	0.02	4.25	0.01
	0.5	3.37	0.03	4.22	0.01
	0.6	0.95	0.39	1.31	0.26
	0.7	3.47	0.02	3.68	0.02
Copper Sulfate	0.2	0.65	0.55	1	0.37
	0.3	0.97	0.39	1.39	0.23
	0.4	0.18	0.87	0.57	0.60
	0.5	1.46	0.22	1.75	0.15
	0.6	1.23	0.28	1.51	0.20
	0.7	4.69	0.01	5.1	<0.01
Diamide	0.2	11.42	<0.01	11.16	<0.01
	0.3	8.59	<0.01	8.69	<0.01
	0.4	4.69	<0.01	4.96	<0.01
	0.5	10.86	<0.01	11.59	<0.01
	0.6	6.01	<0.01	6.09	<0.01
	0.7	1.34	<0.01	1.35	<0.01

## Data Availability

The dataset is obtained from [5] and the source code for the proposed framework is available at Github: https://github.com/shilab/FSF-GA/ (accessed on 16 April 2023).

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
