# Peer review of "FSF-GA: A Feature Selection Framework for Phenotype Prediction Using Genetic Algorithms"

_genes, 2023, doi:10.3390/genes14051059_

Round 1

Reviewer 1 Report

In this manuscript, Shi and Mowlaei proposed a novel feature selection framework for phenotype prediction using a genetic algorithm (FSF-GA). The feature space that required to identify genotypes in predicting a phenotype of interest was greatly reduced using this new model. A yeast dataset was used to conduct extensive experiments for providing a comprehensive vignette study of their method. The results clearly showed that their method has comparable phenotype prediction performance when benchmarked against baseline methods. The selected feature sets can be used to interpret the underlying genetic architecture that contributes to phenotypic variation for the traits under investigation. In summary, this manuscript was well-written, concise, and the simulations and experiments were carefully designed and performed. The significance and novelty were clearly stated that the FSF-GA model as a bi-objective framework for quantitative trait prediction and quantitative trait locus detection. The method is the first to make use of evolutionary algorithms for complicated modeling tasks. Only some minor changes need to be addressed.

Comments:

1)    In line 75, the authors brought “epistasis detection” to the audience the first time, it would be better if the authors could give a brief introduction to this term since it is a new term to general readers.

2)    In Figure 1, how the authors define how many replicates? Why repeated 3 times? For the GA part: “Mutate” meant “Mutations” in genetic sequences?

And does the whole data save to disk before final evaluation? It seems to me that all the backup data were saved before evaluating the final SNP set on the test set.

3)    In Table 1, the authors should specify which type of yeast organism they used. And how did the authors calculate the interaction-to-union ratio? This calculation method was not shown in the “Methods” section.

4)    In Table 2, did the direct overlaps between QTLs detected by FSF-GA and those of previous reports have same SNPs being detected?

5)    In line 195, should be “will be explored”.

66)    In Figure 3 and Figure 5, the authors should specify how many replicates/runs were performed. And it would be great if the authors could show the error bars.

7)    In line 217 to 218, this sentence was not finished. “Show a reasonable….?”

8)    Why the authors think that usage of more advanced models, such as neural networks, can lead to increased prediction performance and better QTL detection? Please give more illustrations on this point.

9)    Using genetic algorithm for feature selection framework of phenotypic prediction is a nice tool, have the authors considered involving more cutting-edge biological techniques? Such as combining CRISPR genetic editing tools and/or base editing tools for more precise SNP studies?

Author Response

We thank the reviewer for the thorough and constructive feedback on our manuscript. We have addressed these comments, and accordingly revised the manuscript and corresponding figures and tables. Below please find our responses to each of the comments in detail (the original comments from the reviewers in black and our responses in red). 

Comments from Reviewer 1  

Point 1: In line 75, the authors brought “epistasis detection” to the audience the first time, it would be better if the authors could give a brief introduction to this term since it is a new term to general readers.  

Response 1: We thank the reviewer for pointing this out. We have revised the manuscript to clarify this, by introducing epistatic effects in line 26-27 and adding another sentence at line 28 to expand on epistasis further. In summary, we have revised the text as the following to clarify this point.  

“These genetic variants can exhibit linear or non-linear interactions, which are respectively referred to as additive or epistatic effects respectively, in determining the trait values. Epistasis represents non-linear relationships among genetic variants and can extend beyond pairwise interactions and encompass higher-order interactions among multiple genetic variants. Consequently, phenotype prediction in presence of epistatic effects is a challenging combinatorial task, since modeling such interactions demands significantly complex models.” 

Point 2: In Figure 1, how the authors define how many replicates? Why repeated 3 times? For the GA part: “Mutate” meant “Mutations” in genetic sequences? 

And does the whole data save to disk before final evaluation? It seems to me that all the backup data were saved before evaluating the final SNP set on the test set. 

Response 2: In Figure 1, the whole process is a single replicate. We repeat the experiment 3 times for the same data split (or replicate) and indicate that with a dashed box and accompanied text “Repeat 3 times, intersect” at the center of the figure. The reason for repeating is that GA is stochastic, and it is not guaranteed to converge to the same answer each time, especially because of the correlations. We chose 3 for our proof of concept, but ideally running the algorithm many times and using the intersection of the results will give us a more confident final set of SNPs. This topic is briefly mentioned in line 265 as well. 

 “Mutate” in genetic algorithm refers to changing the value for at least one parameter in the search space. Here, by “mutate”, we are referring to that of genetic algorithms. The results of the algorithm are kept in memory for the final evaluation. However, we further save them to the disk, alongside the logs for each run for the analysis of the results. 

In the text, to make it clear that the diagram is showing a single run, we have revised the caption for Figure 1 as follows: “The overall pipeline of FSF-GA for a single run. The dataset is split into training and test sets. Pre-processing is then performed on both sets using the information extracted from the training set. Next, GA is applied on the training set three times. The intersection of the results from these three executions is considered as the final set of QTLs and used for the test set evaluation.”. 

Point 3: In Table 1, the authors should specify which type of yeast organism they used. And how did the authors calculate the interaction-to-union ratio? This calculation method was not shown in the “Methods” section. 

Response 3: We have added the type of the yeast organism both in caption of Table 1 and first paragraph of Dataset subsection of Materials and Methods section. There has been a typo in interaction-to-union. We fixed it to “intersection-to-union", and it simply shows how many SNPs intersect vs how many are resulted from a union in 3 GA runs on the same data split. 

We have revised the text as follows:  

Table 1 caption: “SNP Statistics after pre-processing and running the GA three times on the yeast Saccharomyces cerevisiae dataset. LD cutoff refers to the LD threshold used in the pre-processing step. Intersection, in the last column, indicates the overlap between three runs on each LD cutoff and I/U indicates intersection-to-union ratio. Higher values of I/U are desirable and indicate that the results  have more concordance.” 

The first paragraph of Dataset subsection of Materials and Methods section : “In this study, we apply our method to a well characterized yeast Saccharomyces cerevisiae dataset” 

Point 4: In Table 2, did the direct overlaps between QTLs detected by FSF-GA and those of previous reports have same SNPs being detected? 

Response 4: By direct overlaps, we were referring to same SNPs but as you mentioned it was misleading. We have rephrased the caption to better convey the meaning and thank you for mentioning this.  

“Shared QTL SNPs detected by FSF-GA and [32]. Items under overlap positions are represented as chrN_P, where N and P indicate chromosome number and position on the chromosome, respectively. These overlaps show that our method detects relevant features as QTLs.” 

Point 5: In line 195, should be “will be explored”. 

Response 5: We fixed this issue according to your suggestion. 

“Therefore, we expect that using neural networks instead of Bayesian Ridge can lead to a boost in prediction power and better epistatic QTL detection in FSF-GA, subjected to availability of more computational power which will be explored in future work.” 

Point 6: In Figure 3 and Figure 5, the authors should specify how many replicates/runs were performed. And it would be great if the authors could show the error bars. 

Response 6: We have updated the caption for Figure 3 to specify that we run each model for that specific split of dataset only once. The reason is that for 10 traits that we are examining there, we have 6 thresholds and we run FSF-GA 3 times per trait, resulting in a total of 180 runs just for one replicate. We are aware that data splitting can affect the performance of the models, which is why we designed the second experiment in which we use 5-fold cross validation but using 3 traits. The results of the second experiment were previously reported in Figure 5, but as the other reviewer suggested, we replaced it with Table 3 which now contains standard deviations in addition to the average results. Additionally, we added Table 4 which includes the t-test values for the same experiment. 

Caption for Figure 3: “Performance Comparison of FSF-GA to baseline methods based on (a) MAE and (b) MSE (lower values indicate better results) with a one-time evaluation on a single split of the dataset. Α in FSF-GA α corresponds to the LD cutoff threshold used in the pre-processing step. RF and SVM are applied to features for each of six LD cutoff thresholds in pre-processing step, and the best result among them, for each metric, is considered” 

Point 7: In line 217 to 218, this sentence was not finished. “Show a reasonable….?” 

Response 7: Thank you for bringing this to our attention. We fixed this issue. 

“Furthermore, our analysis reveals that QTLs detected by our framework show a reasonable LD concordance with the QTLs identified in previous research. Statistics show that the LD cutoff, in our proposed pre-processing step, affects phenotype predictive power and convergence of the GA in our framework. This LD cutoff is a trade-off between the convergence of the GA and phenotype predictive power of our method.” 

Point 8: Why do the authors think that usage of more advanced models, such as neural networks, can lead to increased prediction performance and better QTL detection? Please give more illustrations on this point. 

Response 8: We have added a sentence to the paper (right after mentioning neural networks in the Discussion section) to elaborate more on this: 

“Some models, such as neural networks and tree-based models, have the capacity to capture non-linear interactions between SNPs. Specifically, neural networks are suitable tools for detecting epistasis since they utilize non-linear activation functions in each layer.” 

Point 9: Using genetic algorithms for feature selection framework of phenotypic prediction is a nice tool, have the authors considered involving more cutting-edge biological techniques? Such as combining CRISPR genetic editing tools and/or base editing tools for more precise SNP studies? 

Response 9: We highly appreciate your suggestion. This application can prove useful, and we have added a brief discussion about this to the first paragraph of the Conclusions section. 

“A practical application of FSF-GA is to combine it with CRISPR genetic editing tools and/or base editing tool for targeted SNP editing and investigating phenotypic changes. This approach could provide precise and targeted SNP studies compared to traditional QTL mapping techniques.” 

Reviewer 2 Report

Phenotype prediction is one of pivotal tasks to understand phenotypic differences introduced by genetic factors. However, it is still an open problem to dissect the genetic contribution to complex phenotypes due to the complexity between genotypes and phenotypes. In view of this, the authors’ motivation is correct and their efforts should be welcome by the science community. But to fit in the increasingly high quality of genes, a compulsory major revision is absolutely needed according to the following points.

(1). The writing of the revised paper should be further improved. The language expression in many places is not standard.

(2). As shown in Fig 1., the whole dataset was divided into 9:1 as the training and test set, and the proportion of test set is too low. It's usually set to a ratio of 8:2, or 7:3.

(3). In the Phenotype prediction section, for the RF regressor, the number of trees were set to p/7. Why the value is set to this value and whether the parameter has been optimized or not.

(4). For performance comparison, the MAE, MSE results were shown in Fig 3. to Fig 5. But it's not very intuitive, it's more objective with tabular data.

(5). The illustrations in the paper are of low quality, e.g., Fig 1., Fig 6.

Author Response

We thank the reviewer for the thorough and constructive feedback on our manuscript. We have addressed these comments, and accordingly revised the manuscript and corresponding figures and tables. Below please find our responses to each of the comments in detail (the original comments from the reviewers in black and our responses in red). 

Point 1: The writing of the revised paper should be further improved. The language expression in many places is not standard.  

Response 1: We asked a native English speaker to help us with improving the language and made changes to rephrase the sentences across the article. 

Point 2: As shown in Fig 1., the whole dataset was divided into 9:1 as the training and test set, and the proportion of test set is too low. It's usually set to a ratio of 8:2, or 7:3. 

Response 2: This point is valid since small test sets may not include challenging samples for prediction. To address this, we have additionally performed a 5-fold cross validation using the first 3 traits, accompanied by the t-Test results which are presented in Tables 3 & 4. 

Point 3: In the Phenotype prediction section, for the RF regressor, the number of trees were set to p/7. Why the value is set to this value and whether the parameter has been optimized or not. 

Response 3: We did not discuss this in the article in detail, but we performed several experiments to fine-tune the RF regressor and p/7 gave us the best results. We were sensitive to get the best results for RF regressor because it was a potential regressor to be used in our GA, but ridge regression gave us better results at the end of the day. We have rephrased the sentence starting at line 141 to briefly mention this fine-tuning. 

“For the RF regressor, after fine-tuning the model, number of trees and maximum depth of each tree are set to ceil(p/ 7) (where p in the number of features) and 7, respectively.” 

Point 4: For performance comparison, the MAE, MSE results were shown in Fig 3. to Fig 5. But it's not very intuitive, it's more objective with tabular data. 

Response 4: You are correct, and we tried to address this issue, but it was only possible to turn Fig. 5 to a table (Table 3). The other two figures have too many columns and rows and we could not fit them in a page. However, to make it possible to have a better look at the data, we have included all our quantitative results in a single excel file as supplementary information which is accessible in the Github repository for our paper: 

 https://github.com/shilab/FSF-GA/blob/main/Supplementary%20files/our_experimental_results.xlsx 

Point 5: The illustrations in the paper are of low quality, e.g., Fig 1., Fig 6. 

Response 5: We have revised the figures with high quality files.  

Round 2

Reviewer 2 Report

  • The paper has been well revised and I have no further questions.
